# Source Identification Analysis of Lead in the Blood of Japanese Children by Stable Isotope Analysis

**DOI:** 10.3390/ijerph17217784

**Published:** 2020-10-24

**Authors:** Mai Takagi, Atsushi Tanaka, Haruhiko Seyama, Ayumi Uematsu, Masayuki Kaji, Jun Yoshinaga

**Affiliations:** 1Department of Environment Systems, University of Tokyo, Chiba 277-8563, Japan; yoshinaga@toyo.jp; 2National Institute for Environmental Studies, Ibaraki 305-8506, Japan; tanako@nies.go.jp (A.T.); seyamah@nies.go.jp (H.S.); 3Division of Endocrinology and Metabolism, Shizuoka Children’s Hospital, Shizuoka 420-8660, Japan; ayumi-uematsu1@i.shizuoka-pho.jp; 4Shizuoka Public Health Center, Shizuoka 420-0846, Japan; kaji_ce@city.shizuoka.lg.jp; 5Faculty of Life Science, Toyo University, Gunma 374-0193, Japan

**Keywords:** lead exposure, lead isotope ratio, Japanese children, source identification, non-dietary source, bioaccessibility

## Abstract

Considering the negative effect of lead (Pb) on children’s neurodevelopment, Pb exposure should be minimized to the lowest extent possible, though the blood Pb (BPb) concentrations in Japanese children are among the lowest in the world. To identify the sources of Pb in blood, isotope ratios (IRs: ^207^Pb/^206^Pb and ^208^Pb/^206^Pb) of Pb (PbIR) in whole blood from eight Japanese children were measured by multi-collector ICP mass spectrometry. Further, samples of house dust, soil, duplicate diet, and tobacco, collected from home environments, were also measured and were compared with PbIR of blood case by case. The relative contribution of Pb in the home environment to BPb were estimated by linear programming (finding an optimal solution which satisfy the combination of IRs and intakes from various sources) when appropriate. Source apportionment for three children could be estimated, and contributions of diet, soil, and house dust were 19–34%, 0–55%, and 20–76%, respectively. PbIR for the remaining five children also suggested that non-dietary sources also contributed to Pb exposure, though quantitative contributions could not be estimated. Non-dietary sources such as soil, house dust, and passive tobacco smoke are also important contributors to Pb exposure for Japanese children based on PbIR results.

## 1. Introduction

In 1991, the Center for Disease Control in the United States (CDC) proposed a blood lead (BPb) concentration of 10 µg/dL as an action level based on findings that Pb affects the cognitive development in children. However, recent epidemiologic studies demonstrate lowered Intelligence Quotient (IQ) and poorer academic achievement at even lower concentrations, i.e., 5 µg/dL or lower [1,2,3,4,5,6]. In addition, Schwartz [7] and Lanphear et al. [8] claimed that no threshold exists for low BPb, where no cognitive effect was observed. Therefore, CDC withdrew the former “level of concern” of 10 µg/dL and set the new intervention level of 5 µg/dL in 2012 [9].

The children’s BPb levels in many countries are currently lower than those measured in the past [10,11,12,13]. However, if no threshold for BPb toxicity exists, further reduction of Pb exposure for children is advisable, even when considering currently low BPb concentrations. To this end, identification of the source(s) of Pb exposure is an essential step.

Historically, sources of Pb contamination in children’s environments were obvious. Air pollution from the use of leaded gasoline [13,14] and the widespread use of Pb-based residential paint [15,16] were major contributors. The former was a global source and affected children directly through inhalation and indirectly by ingestion of contaminated soil, dust, or food [17]. The latter was limited to the countries where such paint was used and affected children directly by ingesting the paint chips and indirectly by contaminating house dust and soil [15,18,19,20,21]. Both leaded gasoline and Pb-based paint have been banned in many countries, and the understanding of contributions from other sources is limited. Therefore, source identification is currently more challenging than in the past.

The use of stable isotope ratios (IRs) of Pb (PbIRs) as indicators is a promising approach for source identification. This approach was used previously for the identification of sources of children in the 1970s and 1980s in the US. These studies confirmed the expected large contributions from Pb-based paint and leaded gasoline during this time [14,15,19,22,23]. A similar approach may allow the identification of current sources, although difficulties are expected since Pb contamination in the environment is typically low, and no obvious major Pb sources exist. PbIRs have been continuously used to evaluate sources of Pb exposure [18,24,25,26]. Such studies indicate that diet or house dust may be current dominant sources for childhood Pb exposure.

In Japan, BPb levels in children are among the lowest in the world [27]. Japan eliminated the use of leaded gasoline before many other countries, and Japanese residences are customarily not painted, which limits or eliminates the historical use of Pb-based paint in homes. Thus, a better understanding of Pb sources is needed for the further reduction in childhood Pb exposure. Two conflicting findings concerning source apportionment for Pb exposure in Japanese children are available. Nakanishi et al. estimated that more than 80% of the oral intake of Pb in Japanese children was from their diet [28], whereas Aung et al. estimated that approximately three-quarters of the oral intake of Pb in children living in the Tokyo metropolitan area was from soil and house dust [29]. Both studies were based on Pb concentrations and ingestion and inhalation rates of exposure media. The other sources of information regarding Pb and the exposure route for Japanese children are limited.

A sensitive analytical method for the measurement of PbIRs in blood and environmental samples has been developed using multi-collector ICP mass spectrometry [30]. This method was employed to identify proximate sources of Pb. Eight children from the Shizuoka Prefecture and Tokyo regions of Japan were assessed on a case-specific basis.

## 2. Materials and Methods 

### 2.1. Subjects and Materials

The children in this study were among the subjects in a previous study on measuring BPb levels in Japan [27]. In that study, blood samples from 352 children in three regions of Japan (Tokyo, Shizuoka Prefecture, and Osaka) were assessed. For children in Shizuoka prefecture and Tokyo (central Japan), doctors asked the caretakers for permission of home visits and environmental sample collection. Twelve caretakers agreed to co-operate this study. They all lived residential areas free of possible Pb contamination from such as metal mining and electric products disposal plant. Four boys (4–7 years) and eight girls (3–11 years) were included. As the potential Pb source sample, soil, vacuum cleaner dust (as house dust), windowsill dust (as outdoor dust) and one-day duplicate diet (duplicate portion of all the foods and drinks they have eaten), were collected. Information on the children and their environment samples is provided in Table 1.

The present study was approved by the Ethical Committee of the University of Tokyo (2008-4R). We got written consent from participants.

### 2.2. Collection of Environmental Samples from Each Home Environment

Blood sampling and home visiting took place during 2006–2010. Sampling kits for environmental samples were provided to caretakers before the day of sampling. The sampling kit contained plastic bags and containers as well as written instructions for sampling duplicate diet and vacuum cleaner dust. All the plastic containers for duplicate diet sampling were cleaned of any Pb contamination by soaking in 6 mol/L nitric acid (HNO_3_) for one week, followed by vigorous rinsing with Millipore water. Oral instruction was also provided during the visit. Caretakers also filled out a questionnaire to describe the characteristics of the residences, smoking habits of family members and the daily activities of children, including playing, hobbies and hand-to-mouth behavior. On the next day of sampling, we visited the home again to receive the samples and collected samples of soil and windowsill dust. Soil samples were collected by plastic scoop from gardens of participants’ houses, elementary schools, or kindergartens. The dust from windowsill was gathered into plastic bags by a brush. When there was a smoking family member in the household, the same tobacco brand used in the home was purchased.

### 2.3. Pretreatment of Exposure Media for Pb Analysis

In the laboratory, the soil, vacuum cleaner dust, and outdoor dust were air-dried to a constant weight. Samples were sieved through a 250 µm stainless-steel screen. The <250 µm fraction was selected since previous study reported that most of soil particles adhered to children’s hands are less than 250 µm [31]. 

Duplicate diet samples, including drinking water, were weighed in one lot for each child. Each sample was individually mixed and homogenized in a single lot using a food processor with a titanium blade and plastic jar (Cuisinart, San-ei Co., Ltd., Tokyo, Japan). For contamination control, the jar was soaked in a dilute HNO_3_ bath for at least two days and rinsed with pure water several times. The blade was washed vigorously with water. Subsequently, 500 mL of dilute ultrapure acetic acid (TAMAPURE, Kawasaki, Japan) was added to the jar, and the food processor operated for 30 s. Acetic acid was removed and analyzed for Pb with ICP mass spectrometry. When Pb was detected (>0.006 ng/mL), washing was repeated by using fresh dilute acetic acid. The duplicate diet sample was then homogenized in the food processor after confirming that no Pb was detected in the acetic acid. An aliquot of homogenized duplicate diet was taken in an acid-washed PFA tube and freeze-dried for approximately one week.

### 2.4. Extraction and Digestion of Environmental Samples

To compare isotope composition of Pb in children’s blood and composition in environmental samples, the bioaccessible fraction of Pb in environmental samples was needed. Bioaccessibility was assessed using a testing method for soil [32] along with analysis for total Pb content. Briefly, the samples were shaken with simulated gastric juice (0.4 mol/L glycine adjusted to pH 1.5 with hydrochloric acid) at 37 °C and were analyzed for Pb concentration (bioaccessible Pb) and IRs. Total Pb in air particulates and tobacco smoke will reach the human alveoli intact; therefore, bioaccessible Pb was not analyzed for outdoor dust. 

Analysis of total Pb concentrations in environmental samples was performed after total digestion with a mixed acid (HNO_3_/perchloric acid/hydrofluoric acid). Blood and tobacco leaves were digested with HNO_3_ by the “double vessel method” [33]. Water purified with MilliporeR system and ultrapure grade acids (TAMAPURE, Kawasaki, Japan) were used throughout the study and other reagents (e.g., glycine) were used after checking for Pb contamination.

### 2.5. Pb Analysis

Bioaccessible Pb or acid digested Pb in environmental samples and acid digested Pb in blood were separated and purified by anion exchange using a Pb bromide complex [30]. The separated fraction was made up in 0.14 mol/L HNO_3_ solution. All the separation procedures were carried out in a class 1000 clean room.

IRs of Pb (^207^Pb/^206^Pb and ^208^Pb/^206^Pb) were determined by multi-collector ICP mass spectrometry (IsoProbe, Thermo Fisher Scientific, MA) equipped with Aridus II membrane desolvation system (CETAC, Omaha, USA) using thallium (SRM 997, National Institute of Standard and Technology) as an external standard for mass bias correction. Detailed condition of MC-ICPMS is described in Takagi et al. (2011) [30]. The typical precision of the IR measurement (two times the standard deviation) was both 0.02%for ^207^Pb/^206^Pb and ^208^Pb/^206^Pb [30]. 

Pb concentration in bioaccessible fractions, acid-digested solutions, and purified fractions for IR analysis were determined by ICP quadrupole mass spectrometry (Agilent 7500ce, Agilent Technologies Co. Ltd., Tokyo, Japan). PbIRs in environmental samples and blood determined by the present method were validated by the analyses of certified reference materials, i.e., NIST SRM 2709, San Joaquin soil, NIST SRM 2711, Montana Soil, NIES CRM NO.27, Typical Japanese Diet (National Institute for Environmental Studies, Japan) and SeronormTM Trace Elements Whole Blood L-1 (Sero, Norway) with reported IRs and concentrations [30].

### 2.6. Estimation of Daily Pb Exposure by the Conventional Method

Before source apportionment analysis using IR, daily Pb exposure levels were estimated from total Pb concentrations in the environment samples by conventional methods. The sum of products of Pb concentration and ingestion/inhalation rates was calculated for each environment sample for each subject. Subjects E, J, K, and L were excluded because not all environmental samples were collected. When the contribution of a certain exposure medium was expected to be low (within 2%), the medium was excluded from the source apportionment analysis. Ingestion rates for diet were measured in duplicate for one day. Ingestion rates for soil and house dust were estimated to be 8.5 and 25 mg/day, respectively. These ingestion rates are based on a soil ingestion study by the Ministry of Environment of Japan, based on input/output balance of aluminum in 33 children [34]. The excess aluminum output was 1.4 mg/day. We assumed that the excess aluminum output was from both house dust and soil at a ratio of 1:1. The average aluminum concentrations in house dust and soil were 2.5% and 8%, respectively [35,36]. Results from this study were used to calculate house dust ingestion rate, (0.7 mg/day)/(0.025 g/g) = 28 mg/day, which was rounded to 25 mg/day, and soil ingestion rate, (0.7 mg/day)/(0.08 g/g) = 8.7 mg/day, which was rounded to 8.5 mg/day.

Inhalation exposure to Pb was estimated using Pb concentrations in outdoor dust (mg/kg), monitoring data on atmospheric concentration of suspended particulate matter measured at the nearest monitoring station (µg/m^3^), and the default inhalation rate of children (9.9 m^3^/day) [36]. In this conventional calculation, contributions from tobacco were not included because the passive exposure of children could not be defined quantitatively.

### 2.7. Calculation of the Source Apportionment

In the present study, “uptake” is defined as intake of the bioaccessible fraction. Amounts of uptake (µg/day) of Pb from unknown factors in the present study, i.e., soil, house dust, and passive tobacco smoke, were estimated assuming that the sum of the IRs weighted by the amount of uptake from each source are consistent with PbIR in blood within two standard deviations (SDs) (Equation (1)).
B_IR –_ 2SD ≤ S_IR_ × a/D + H_IR_ × b/D + T_IR_ × c/D + F_IR_ × z/D ≤ B_IR_ + 2SD(1)
where BIR, SIR, HIR, and FIR are IRs of blood, soil, house dust, and diet, respectively. a, b, and c are uptake from soil, house dust, and passive tobacco smoke, respectively. z is the amount of Pb uptake from the diet, which is calculated by the bioaccessible Pb concentration of diet samples and measured weight of the total diet. D is amount of daily Pb uptake estimated by BPb concentration and a conversion factor reported by FAO/WHO Joint Expert Committee on Food Additives (JECFA). JECFA used the factor “0.16 µg/dL of BPb concentration equals to 1 µg/day of daily intake” when setting the Provisional Tolerable Weekly Intake (PTWI) of Pb [37]. The factor was set assuming that all Pb intakes are from the diet, and daily uptake was calculated using daily dietary intake and bioaccessibility of Pb in the diet (0.68). The bioaccessibility was calculated as the average of bioaccessibility of Pb in diet samples in the present study and that reported by Aung et al. [29]. For example, when the BPb concentration is 1.0 µg/dL, estimated daily Pb uptake (D) is 4.25 µg/day (1.0 ÷ 0.16 × 0.68). Estimated Pb uptake based on BPb concentrations for each child is shown in Table 2. Linear programming method [38] were used to estimate the range of the a, b and c using the solver tool in Excel 2013. “Pb intake” via diet, soil, and house dust was estimated by “Pb uptake” via diet, soil, and house dust and their respective bioaccessibility. We assumed that the bioaccessibility of Pb from tobacco smoke is 100%.

## 3. Results

### 3.1. Lead Concentrations in Environment Samples

Total and bioaccessible Pb concentrations in environmental samples from 12 households were obtained and analyzed (Table 3). Full sets of environmental data were collected from eight households. The remaining other four included two households with an incomplete collection of duplicate diet samples (subject J and L), one household with a missing soil sample (subject E), and one missing all samples, except vacuum dust (subject K). Brief information about each environmental sample is provided (Table 1). The number of samples is not equal to 12 because some samples were missing, or multiple samples were collected from some households (Table 3). Note that Pb concentration in this table is expressed as a unit weight of each environment sample (air-dried or fresh weight) that is the most likely actual exposure conditions. The rank of average total Pb concentration is diet < tobacco (leaves) < soil < house dust < outdoor dust. Similar mean bioaccessibility was soil < house dust < diet.

### 3.2. Estimated Daily Pb Intakes of Children Using Conventional Methods

Daily Pb exposures were calculated using measured total Pb concentrations and default ingestion/inhalation rates, except for diet where the actual measured ingestion rate (g/day) was used. The mean ± SD ingestion of the duplicate diet was 1547 ± 595 g, which was comparable to the value of a Japanese total diet study (1796 g, 2017, 7–14 years old, [39]). Estimated daily Pb intake of the eight children, estimated with the conventional method, ranged from 2.4 to 13.9 µg/day (mean ± SD, 6.1 ± 4.6 µg/day). Estimated contributions of each medium (minimum–maximum) were 1.8%–6.8%, 11%–78%, 18%–87%, and 0.073%–2.0%, for soil, house dust, diet, and air, respectively, based on the conventional method.

### 3.3. IRs of Pb in Blood and Environmental Samples

PbIRs (^207^Pb/^206^Pb and ^208^Pb/^206^Pb) of the total Pb in the blood and tobacco leaves and those in bioaccessible fractions of environmental samples for eight children (A–D and F–I) were measured by MC-ICPMS (Table 4) and plotted (Figure 1). Error bars are not shown because they are obscured by marker symbols. The contribution of outdoor dust was less than 2% and this source was excluded from the calculations. The mean ^207^Pb/^206^Pb ratios, in the lowest to the highest order, were tobacco leaves < soil < diet < blood < house dust (Table 4, Figure 1). Variation of PbIRs among soil samples was large, whereas that among the house dust and blood samples were relatively small. Pb in blood and house dust reflects a mixture of sources with varying IRs, which may explain the relatively small variation of PbIRs in these media.

Some tobacco samples for subjects B, D, F, and I (two of three) were excluded from the calculations (gray colored in Figure 1) because household questionnaires indicated that smoking did not occur when children were present.

Two standard deviations error bars (2SD) were obscured by marker symbols. Gray-colored IRs of tobacco (+) in subjects B, D, F, and I were not included in the linear programming analysis because the tobacco was not used in rooms when children were present.

### 3.4. Source Identification Based on PbIR

Linear programming analysis was used to estimate apportionment of each exposure medium for six children (subjects A, B, C, D, G, and H). For subjects F and I, estimated total uptake via diet (bioaccessible Pb concentration in diet × weight of the duplicate diet) was larger than uptake estimated from BPb concentrations (Table 2). Data for these children were excluded from source apportionment calculations. For subjects B, C, and D, a viable solution was reached. Results of “Pb uptake” via soil (a in Equation (1)) and house dust (b in Equation (1)) and “Pb intake” via soil (a/bioaccessibility of soil) and house dust (b/bioaccessibility) of the three subjects are shown in Table 5. Soil and house dust were primary sources for Pb uptake for subject B; house dust was primary sources for subject C; and all sources, diet, soil, and house dust contributed comparably for subject D. A linear programming solution could not be reached for subject A, G, and H. 

## 4. Discussion

### 4.1. Pb Concentration in Environmental Samples

Total concentrations of Pb in soils, house dust, outdoor dust, and diet in the present study (Table 3) were similar to concentrations reported in previous studies in Japan [29,40,41,42]. Aung et al. [29] demonstrated that the mean bioaccessibility of Japanese soil, house dust, and diet were 43% (n = 44), 57% (n = 20), and 52% (n = 6), respectively, measured with a method similar to that used in the present study. Mean bioaccessibilities of soil (35%) and house dust (57%) in this present study were comparable to values reported by Aung et al. [29], but the bioaccessibility of Pb in the diet (87%) was larger. The reason for this difference is unclear, the small sample size of both studies could contribute.

### 4.2. Source Identification of Pb in Children

Estimated daily Pb intakes of the eight children by the conventional method (2.4–13.9 µg/day, 9.68–2.4 µg/kg/week; calculated by each body weight) were much smaller than the former PTWI established by WHO-JECFA (25 µg/kg/week [37]). However, this PTWI has been withdrawn to reflect the finding of neurodevelopmental effects at lower intake levels [43]. Using the conventional method, contributions of outdoor dust were negligible (less than 2% of total daily intake). Outdoor dust was, therefore, excluded from IR analysis. High-precision isotopic analysis by MC-ICPMS was able to distinguish PbIRs in environmental samples, and the case-specific approach showed that PbIRs of blood and environmental samples distributed differently for each child (Figure 1). Generally, PbIRs of blood distributed on the line drawn between those of house dust and diet (subject B, C, D, G, H, and I), indicating that Pb in blood mainly originated from house dust and the diet.

Source apportionment for three subjects (Table 5) suggested that exposures via soil and house dust were primary sources for BPb. This result is consistent with Aung et al. in which predominant contributions from the soil and house dust were identified among Japanese children, by using conventional methods. Contributions of house dust to total Pb in blood (10%–87% of Pb intake, Table 5) were comparable to those estimated by our conventional method (11%–78%), but the contributions of soil based on PbIR (0%–79% of Pb intake, Table 5) were higher than those estimated by our conventional method (2%–7%). The IR method indicated that soil ingestion may be greater than the default value of 8.5 mg/day for these children.

The present case study results were not consistent with the previously estimated source-specific contributions of Pb in Japanese children by a probabilistic approach, in which 80% of daily intake was from the diet [28]. This inconsistency might be due in part to the use of a case-specific approach used in the present study. However, dietary intake data used by Nakanishi et al. (24 µg/day) were considerably higher than the currently reported levels estimated by 24-h duplicate diet study (mean: 4.8 µg/day (n = 33) [29]; geometric mean: 2.28 µg/day (n = 296) [44]). Nakanishi et al. [28] used the data of market basket study conducted during 1999–2003 by National Institute of Health Sciences [45]. We cannot identify the clear reasons of the difference, but we suspected that possible reasons were due to differences in year or place of survey.

For three subjects (A, G, and H), linear programing could not find the solution. For subject A, PbIR of blood was slightly outside of the area encompassed by PbIRs of environment samples (Figure 1A). PbIR of blood was close to that of house dust, and this medium could be the major source of Pb for this child. Variability of PbIR of the house dust samples might be the reason for the slight deviation of the blood PbIR, that led to the failure of linear programming. Notably, the BPb concentration for this subject was higher (3.1 µg/dL) than those of other subjects, even though Pb concentrations in environmental samples collected from this subject’s home were not significant compared with the concentrations found in the environments of the homes of other subjects. The presence of other Pb source(s) for this child that was not sampled, e.g., stationaries or toys, could have affected BPb.

For subject H, PbIRs of blood was slightly outside the small area encompassed by PbIRs in house dust, diet, and two soils (Figure 1H). The PbIRs of environmental samples were closer to each other than other subjects’ environmental samples, and this observation might be a reason that linear programming failed. Since PbIR from each environmental sample may contain inherent small variations, e.g., day to day, the relative magnitude of intra-source variation could be much larger than inter-source variation that could result in mathematical instability. Apart from a quantitative estimation, the figure indicates that the diet, soil, and house dust contributed approximately to blood Pb. 

For subject G, PbIR of blood located fell between the IRs of house dust, soil, and diet. Still, linear programming could not find a solution. A variation in PbIRs of the environmental samples was as small as that of subject H, and the same reason for a failure in linear programming solution may be postulated. Of note, the PbIR of blood was more like that of soil and diet than house dust.

In the cases of subjects F and I, linear programming was not applicable because either higher Pb uptake via diet was found than estimated total lead uptake calculated based on BPb, or z/D in Equation (1) was more than 1. When z/D is larger than 1, a and b cannot meet the requirements of a > 0 and b > 0 in Equation (1). Potential day-to-day variation of Pb concentration in the diet could result in z/D > 1. PbIRs of blood for subject F were less than the ratios in soil, house dust, and diet, indicating that tobacco smoke significantly contributed to BPb. Smoking in the same room with subject F did not occur according to the questionnaire. However, the PbIRs do not exclude the possibility of passive exposure for this subject. Yoshinaga et al. [27] reported that passive smoking had a significant effect on the BPb of Japanese children. Passive smoking is also one of the sources of Pb exposure in US children [46,47]. For subject I, PbIRs of environmental samples and blood fell along the same line (Figure 1I). In such case, it is difficult to quantify apportionment.

### 4.3. Limitation and Uncertainty of This Study

This study was conducted using a case-specific approach for only eight children in Shizuoka Prefecture and Tokyo and samples were collected in late 2000s. Future studies with current approach for a larger number of children and other areas in Japan are needed to obtain a more general picture of Pb source apportionment. Other limitations and uncertainties include: (1) the conversion of BPb concentration to daily intake or uptake of Pb, (2) time-lag between blood sampling and environmental sample collection, (3) representativeness of samples, and (4) the presence of other Pb source(s).

First, we used the conversion factor (from BPb concentration to amount of Pb uptake) of infants and children which was cited in [37]. According to the report, the conversion factor was empirically derived from only one study. Thus, uncertainty, which is inherent in this factor, is not known. Improved knowledge of the pharmacokinetics of Pb will reduce this uncertainty in this source apportionment analysis. Second, Pb in the blood reflects Pb intake during 1–2 months before blood sampling because the biological half-life of BPb is 28–36 days [48]. In the present study, environmental samples were collected several months after blood sampling. Third, one house dust sample and one-day duplicate diet from each child may be not enough for representativeness. However, we assumed a small variability in the PbIRs of environmental samples because Pb sources, such as tap water or container to diet, and outdoor dust or paints to house dust and soil, are not likely to change significantly over short periods in-home environments. Accidental mixing of Pb from routine sources may occur more often than we anticipated. However, this assumption may not always be accurate. More data of in-home variability of Pb concentration and PbIR in diet and house dust is necessary to support this assumption in future studies. Forth, if an unexpected source(s) exists, source apportionment with the present approach will be insufficient even though typical possible sources were considered. When the PbIR of blood cannot be explained by PbIRs of typical sources, a further study to identify additional sources is needed.

## 5. Conclusions

Pb concentrations and PbIRs in blood and environmental samples from each home environment were measured to identify the sources of Pb exposure for Japanese children. We could demonstrate the valuable examples to estimate source apportionment of Pb in children who had very low BPb concentration using PbIR. In addition, this study is the first in Japan to identify the sources of Pb in children using PbIR analysis. We found that there were the cases that non-dietary sources such as soil, house dust, and passive smoking had also non-negligible contributions to Pb exposure. From these results, to reduce Pb exposure of Japanese children, reduction of exposure via soil, house dust, and passive tobacco smoke could be important for some cases. Further studies involving more children and other areas in Japan, using IR, are needed.

## Figures and Tables

**Figure 1 ijerph-17-07784-f001:**
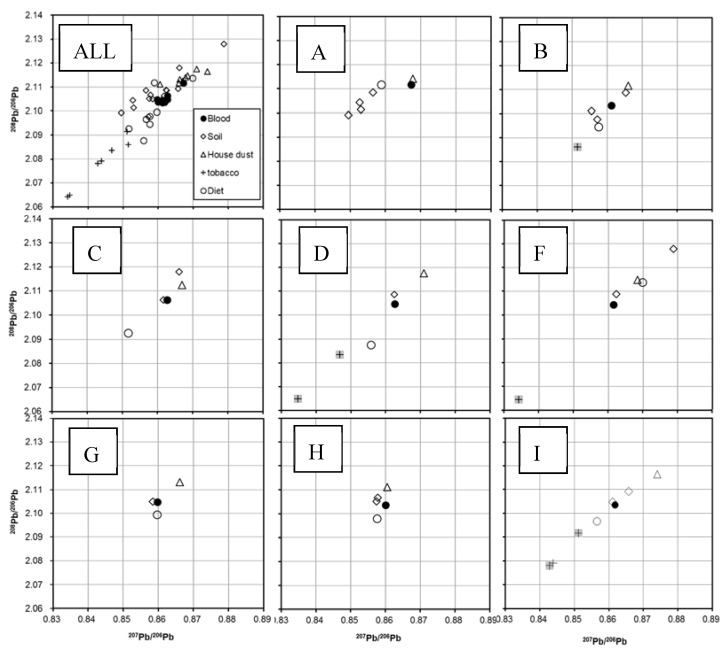
Lead isotope ratios (^207^Pb/^206^Pb, ^208^Pb/^206^Pb) of all subjects (**ALL**) and of individual subjects (**A**–**I**).

**Table 1 ijerph-17-07784-t001:** Characteristics of study subjects (n = 12).

ID	Age	Sex ^a^	Blood Pb (µg/dL)	Sampling Date	Residential Area	Collected Environmental Samples ^b^
A	7	M	3.1	August, 2006	Shizuoka	S, H, D, F, To
B	8	F	1.1	August, 2006	Shizuoka	S, H, D, F, To
C	8	F	1.5	April, 2009	Shizuoka	S, H, D, F
D	7	M	0.46	May, 2009	Shizuoka	S, H, D, F, To
E	9	F	1.1	September, 2009	Shizuoka	H, D, F, To
F	11	F	1.3	December, 2009	Shizuoka	S, H, D, F, To
G	6	F	0.62	January, 2010	Shizuoka	S, H, D, F
H	4	M	0.52	February, 2010	Shizuoka	S, H, D, F
I	9	F	1.1	March, 2010	Tokyo	S, H, D, F, To
J	3	F	0.99	February, 2010	Shizuoka	S, H, D, F ^c^, To
K	3	M	1.5	May, 2010	Tokyo	H
L	9	F	3.3	September, 2006	Shizuoka	S, H, D, F ^c^, To

^a^ M: Male, F: Female; ^b^ Letters indicate kinds of environmental samples: S (Soil), H (House dust), D (Outdoor dust), F (total diet), and To (Tobacco); ^c^ Duplicate diet samples were incomplete.

**Table 2 ijerph-17-07784-t002:** Estimated Pb uptake calculated from blood Pb concentration.

ID	Pb Uptake ^a^ (µg/day)
A	13.2
B	4.25
C	6.29
D	1.96
F	5.67
G	2.65
H	2.19
I	4.48

^a^ Amount of Pb uptake was calculated as follows: BPb (µg/dL) ÷ 0.16((µg/day)/(µg/dL)) × bioaccessibility of diet (0.68). Bioaccessibility of diet was based on Aung et al. (2004) and this study.

**Table 3 ijerph-17-07784-t003:** Pb concentrations in environmental samples collected from subject households.

Sample	n	Total Concentration ^a^ (Min–Max) mg/kg ^b^	Bioaccessible Concentration ^a^ (Min–Max) mg/kg ^b^	Bioaccessibility ^c^ (Min–Max) %
Soil	17	24.2 ± 18.7 (8.7–80.3)	9.9 ± 10.7 (0.7–41.4)	34.6 ± 13.4 (5.5–52.6)
House dust	12	69.7 ± 47.0 (24.0–155)	34.3 ± 24.8 (12.3–102)	56.9 ± 21.6 (7.7–96.3)
Outdoor dust	13	165 ± 141 (39.8–529)	NA ^d^	NA ^d^
Duplicate diet	9	0.0022 ± 0.0017 (0.0007–0.0065)	0.0020 ± 0.0019 (0.0007–0.0063)	87.2 ± 13.6 (66.0–100)
Tobacco leaves	10	0.81 ± 0.23 (0.56–1.37)	NA ^d^	NA ^d^

^a^ Mean ± SD; ^b^ Concentration is on an air-dried basis for soil and dust and oven-dried basis for outdoor dust. Dietary concentration is expressed on a fresh weight basis; ^c^ Bioaccessibility = [(bioaccessible Pb concentration)/(total Pb concentration)] × 100; ^d^ Bioaccessible Pb was not analyzed because all fraction of Pb in outdoor dust and tobacco leaves is assumed to be absorbed.

**Table 4 ijerph-17-07784-t004:** Isotope ratios of blood Pb and bioaccessible Pb in environment samples.

Samples	n	^207^Pb/^206^Pb Mean ± 2SD	^208^Pb/^206^Pb Mean ± 2SD
Blood	8	0.8621 ± 0.0047	2.1053 ± 0.0055
Soil	15	0.8600 ± 0.0140	2.1074 ± 0.0148
House dust	8	0.8676 ± 0.0079	2.1139 ± 0.0045
Duplicate diet	8	0.8584 ± 0.0052	2.0993 ± 0.0181
Tobacco leaves ^a^	7	0.8436 ± 0.0141	2.0782 ± 0.0207

^a^ IRs of total Pb.

**Table 5 ijerph-17-07784-t005:** Range of source apportionment estimates from linear programming.

Media		Pb Uptake ^a^		Pb Intake ^b^	
		µg/day	Percentage	µg/day	Percentage
B	Diet	0.81	19%	1.2	8.6%–12%
	Soil	1.1–2.1	26%–49%	5.7–11	57%–79%
	House dust	1.4–2.4	32%–55%	1.4–3.2	10%–32%
	Total	4.25	100%	10–14	100%
C	Diet	1.5	24%	1.6	15%
	Soil 1	0–0.84	0%–13%	0–2.3	0%–21%
	Soil 2	0–0.94	0%–15%	0–1.8	0%–16%
	House dust	3.9–4.8	61%–76%	7.3–9.1	66%–83%
	Total	6.29	100%	11	100%
D	Diet	0.68	34%	0.88	5.9%–11%
	Soil	0.60–0.93	30%–46%	1.5–2.3	10%–28%
	House dust	0.39–0.70	20%–36%	4.9–13	60%–87%
	Total	1.96	100%	8.1–15	100%

^a^: Pb uptake was calculated by linear programming; ^b^: Pb intake was calculated from Pb uptake via each medium and the measured bioaccessibility in each.

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
