# Peer review of "Source Identification Analysis of Lead in the Blood of Japanese Children by Stable Isotope Analysis"

_ijerph, 2020, doi:10.3390/ijerph17217784_

Round 1

Reviewer 1 Report

This manuscript deal with the source apportionment of lead in 8 Japanese children blood, using chemical results from different lead isotopes. This issue is important, and the research is very interesting.

I really enjoyed reading this paper.

I recommend accepting as is.

My only observation is in Page 1 Line 38: I do not know if 5 μg/dL should be 5 mg/dL

Author Response

We really appreciate your valuable comment. The response is as follows:

Point 1: Page 1 Line 38: I do not know if 5 μg/dL should be 5 mg/dL.

Response 1: The 5 μg/dL is correct. In the review manuscript, most “μg“ were changed to ”mg”, therefore, we corrected all appropriately.

Reviewer 2 Report

MS No: ijerph-909650

MS Title: Source Identification Analysis of Lead in the Blood of Japanese Children by Stable Isotope Analysis

Article Type: Original Research

Takagi et al., aim to identify the sources of Pb in whole blood, isotope ratios (IRs: 207Pb/206Pb and 208Pb/206Pb) of Pb (PbIR), from eight Japanese children by multi-collector ICP mass spectrometry. They also analyzed the PbIR of those samples in house dust, soil, duplicate diet, and tobacco, collected from home environments and were compared the results with PbIR of blood case-by-case. The relative contribution of Pb in the home environment to BPb were estimated by linear programming when appropriate. Source apportionment for three children could be estimated, and contributions of diet, soil, and house dust were 19%–34%, 13%–55%, and 20%–76%, respectively. The PbIR for the remaining five children suggested that non-dietary sources also contributed to Pb exposure. Eventually, they suggested non-dietary sources such as soil, house dust, and passive tobacco smoke might be also important contributors to Pb exposure for Japanese children based on PbIR results. My major concern is that the authors analyze those samples from the very limited cases but aim to tell a great story. Additionally, there are several typo errors and the authors should carefully recheck if they have revision opportunity. Otherwise, PbIR determined by multi-collector ICP mass spectrometry remains a good tool to retrospectively analyze those samples with low Pb concentrations collected from many years ago.

Author Response

We really appreciate your valuable comments. We tried to respond point-by-point to the comments. Please see the attachment.

Reviewer 3 Report

This is an interesting paper on a relevant subject but with very few subjects of which more than half have too few data to draw reliable conclusions which renders the study an interesting case study at best but doesn't really allow to make any firm conclusions. The data can be used though to inform new research projects.

One of the major concerns is on the validation of the applied methodology for source apportionment. The methodology is described but there is no information on how the authors know that it actually works out well and the number of cases is very small. 

Minor comments:

  •  explain 'duplicate diet' in more detail
  •  line 42-43: duplication
  •  line 112: '< 250 mm' must be an error
  •  line 162: what is the basis for the expectation ?

Author Response

(The authors gave the same response as above.)

Reviewer 4 Report

This is an interesting study about a topic of interest to environmentalist and pediatric neurologists. The paper is well written and the methods are written with suitable levels of detail. The results are clear and the discussion seems appropriate.  My one concern was that there was no discussion about the retrospective nature of the study and long time lag between the years of sampling and now. Although listed as a limitation, it deserves more discussion, Can the authors comment on the justification for a 14 year lag, during which time the air quality must surely have changed. 

Author Response

(The authors gave the same response as above.)

Round 2

Reviewer 2 Report

My major concern remains the same that the very limited cases limits the novelty and attractions for the readers. The second one is the serious ethical concern that they should obtain the agreement from those cases collected on 2002 if they do different assays for different new applications or purposes. Otherwise, these cases had previously agreed that their specimens could be used for freely any purpose. Can they also provide something else to be positive controls since the specimens have been in their hand for 18 years?

Author Response

Thank you for the comments.

Mai Takagi
